# Neural network approach to reconstructing spectral functions and complex poles of confined particles

**Thibault Lechien[1\*] and David Dudal[2,3†]**

**1** Department of Computer Science, KU Leuven Campus Kortrijk - Kulak,
Etienne Sabbelaan 53, Kortrijk, 8500, Belgium
**2** Department of Physics, KU Leuven Campus Kortrijk - Kulak,
Etienne Sabbelaan 53, Kortrijk, 8500, Belgium
**3** Ghent University, Department of Physics and Astronomy,
Krijgslaan 281-S9, Ghent, 9000, Belgium

⋆ thibault.lechien@student.kuleuven.be , † david.dudal@kuleuven.be

## Abstract

Reconstructing spectral functions from propagator data is difficult as solving the analytic continuation problem or applying an inverse integral transformation are ill-conditioned problems. Recent work has proposed using neural networks to solve this problem and has shown promising results, either matching or improving upon the performance of other methods. We generalize this approach by not only reconstructing spectral functions, but also (possible) pairs of complex poles or an infrared (IR) cutoff. We train our network on physically motivated toy functions, examine the reconstruction accuracy and check its robustness to noise. Encouraging results are found on both toy functions and genuine lattice QCD data for the gluon propagator, suggesting that this approach may lead to significant improvements over current state-of-the-art methods.

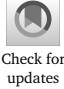

# 1 Introduction

For non-perturbative approaches to Quantum Field Theory, determining the analytical structure of Euclidean space correlation functions is difficult. Real physics is happening in Minkowskian space, but for computational reasons, one is frequently forced to work in Euclidean space, thinking in particular about lattice Monte Carlo simulations which require a positive weight function [1]. The analytical continuation of these Euclidean functions into the complex plane towards Minkowskian space becomes a harder problem than for perturbative approaches, where one can rely on the Wick rotation.

A lattice Monte Carlo Euclidean propagator is only available in a discrete form and it is well-known that the analytical continuation is only unique when departing from a function over an open subset of $\mathbb{C}$. For two-point correlation functions, the information we want to access is encoded in the spectral density $\rho(\omega)$, which is linked to the Euclidean propagator by the Källén-Lehmann spectral representation

$$D(p) = \int_{\sigma}^{\infty} \frac{\rho(\omega)}{\omega^2 + p^2} d(\omega^2), \tag{1}$$

with $p_\mu$ the momentum variable and $\sigma$ an IR cutoff, possibly zero. Strictly speaking, the foregoing spectral representation can be derived on quite general grounds for at least physical (observable) particles, in which case it necessarily holds that $\rho(\omega) \geq 0$ [2].

In strongly coupled non-Abelian gauge theories, with Quantum Chromodynamics (QCD) as the archetypical example realized in nature, one encounters the phenomenon of confinement [3, 4]: the elementary particles are no longer part of the physical spectrum, but appear in bound states, for example a proton built from 3 quarks. Unfortunately, there is far less known about the possible spectral properties of these confined particles.

Traditionally, reconstructing spectral functions from propagator data can be done through e.g. Bayesian inference [5], of which a variant is the Maximum Entropy Method [6]. It regularizes the inversion by incorporating prior domain knowledge on the shape of the spectral function, including imposing positivity. Another approach to obtain the Källén-Lehmann spectral density is given in Dudal et al. [7], which regularizes the problem by implementing a version of Tikhonov regularization supplemented with the Morozov discrepancy principle. This can also cover the case of non-positive $\rho(\omega)$, while positivity can be imposed via an appropriate constraint minimization [8], see [9] for a concrete application. Another option is to directly solve, in a approximative fashion, the quantum equations of motion in the complex momentum plane [10–12].

In this paper we focus on a generalized spectral representation for the propagator, where pairs of complex conjugate simple poles (Eq. (2)) are also allowed, next to a non-positive spectral density $\rho(\omega)$. Such generalizations have been considered in literature before [13–15] on theoretical grounds, and are exhibited in e.g. the gluon propagator of the massive Yang-Mills model [16], which serves as an effective description of the gluon [17]. Other evidence in favour of such poles can be found in, for example, [11, 18, 19].

When sets of complex conjugate poles are present, the numerical reconstruction problem becomes more difficult. Binosi et al. [20] proposed an inversion method using rational function (Padé) interpolation, see also [21]. Their findings suggest that the presence of the complex conjugate poles is a characteristic of the gluon 2-point function and they were able to reconstruct both spectral functions and poles with reasonable accuracy.

Recently, using supervised machine learning and specifically feedforward neural networks to reconstruct spectral functions from propagator data has seen more attention thanks to its superior performance. Examples are Fournier et al. [22] and Yoon et al. [23] which both trained neural networks to perform analytic continuation on normalized sums of Gaussian

distributions. Kades et al. [24] and Wang et al. [25] used linear combinations of unnormalized Breit-Wigner peaks as their mock spectral functions. In comparison, we propose a more intricate spectral function and reconstruct not only the spectral function, but also pairs of complex poles and an IR cutoff, generalizing this approach and as such considerably extending its potential applicability.

The remainder of this paper is structured as follows. In Section 2 the general spectral representation is given and our method of data generation is illustrated. In Section 3 we explain the architecture and workings of our neural network. Section 4 examines the quality of the reconstruction, its robustness to noise and how well our method works on genuine lattice QCD data. Finally, a conclusion and possible avenues for further research are explored in Section 5.

## 2 General spectral representation

We consider a generalized Källén-Lehmann spectral representation in the presence of $n$ complex conjugate pairs of simple poles, located at $q_i, q_i^*$ ($\in \mathbb{C}$) with residues $R_i, R_i^*$ ($\in \mathbb{C}$) (Eq. (2)). Generally a spectral function can be obtained from the inverse integral transformation or by analytic continuation of its Euclidean propagator through Eq. (3), where $\rho(\omega)$ is the non-positive definite spectral density function and $\epsilon \to 0^+$ [16]. We will attempt to reconstruct the spectral function $\rho(\omega)$, the poles $q_i, q_i^*$, the residues $R_i, R_i^*$ and the potential IR cut-off $\sigma$ from propagator data $D(p^2)$ by training an artificial neural network on toy functions. Concretely, we will consider

$$D(p^2) = \int_\sigma^\infty \frac{\rho(\omega)}{\omega^2 + p^2} d(\omega^2) + \sum_i \frac{R_i}{p^2 + q_i} + \sum_i \frac{R_i^*}{p^2 + q_i^*}, \qquad (2)$$

whereby, thanks to Cauchy's theorem,

$$\rho(\omega) = 2 \operatorname{Im} D(-i(\omega + i\epsilon)). \qquad (3)$$

### 2.1 Data generation

In order to generate toy spectral functions, we used Eq. (4)-(6). We expect $\rho(\omega)$ to be composed of positive and negative peaks with certain widths. Thanks to asymptotic freedom in the ultraviolet[1], we know that asymptotically, $\rho(\omega) \to \frac{-Z}{\omega^2 \ln(\frac{\omega^2 + m^2}{\lambda^2})^{1+\gamma}}$ for $\omega \to +\infty$ and $Z > 0$ [7, 26, 27]. We propose $\rho(\omega) = \rho_1(\omega) + \rho_2(\omega)$, where $\rho_1(\omega)$ are Breit-Wigner forms multiplied by a factor which gives the desired asymptotes and $\rho_2(\omega)$ is a sum of Gaussian distributions and derivatives of such distributions. Note that the addition of $\rho_2(\omega)$ will not change the UV tail for $\omega \to +\infty$. For $\beta_i$ and $\tilde{\beta}_i \to 0$, these are regularized $\delta$- and $\delta'$-functions. The latter were considered as possible generalized spectral ingredients in [28].

We then integrate these spectral functions in Eq. (7) and add pairs of complex conjugate simple poles. The parameters' values can be found in Eq. (8)-(12), where $b_j$ and $d_j$ are non-negative without loss of generality. These parameter ranges were based on previous works' estimates [19–21] and preliminary experiments which found that larger ranges rarely fit the imposed constraints. There are namely four constraints (Eq. (13)-(16)): first that the propagator must be positive, then two constraints that prevent a division by zero and a final one that

---

[1]Meaning that the coupling constant becomes small at high energies, indicating that perturbation theory becomes applicable.

encodes the asymptotic behavior of the spectral function for small frequencies in the presence of complex poles [29].

$$\rho_1(\omega) = \frac{-Z}{\ln(\frac{\omega^2+m^2}{\lambda^2})^{1+\gamma}} \left( \sum_{i=1}^{N_1} \frac{A_i \omega^2}{B_i \omega^2 + (C_i - \omega^2)^2} \right), \tag{4}$$

$$\rho_2(\omega) = \sum_{i=1}^{N_2} \gamma_i e^{-\frac{(\omega^2-\alpha_i)^2}{\beta_i}} + \sum_{i=1}^{N_3} \tilde{\gamma}_i \omega^2 e^{-\frac{(\omega^2-\tilde{\alpha}_i)^2}{\tilde{\beta}_i}}, \tag{5}$$

$$\rho(\omega) = \rho_1(\omega) + \rho_2(\omega), \tag{6}$$

$$D(p^2) = \int_\sigma^\infty \frac{\rho(\omega)}{\omega^2+p^2} d(\omega^2) + \sum_{j=1}^{N_4} \frac{a_j + ib_j}{p^2 + c_j + id_j} + \sum_{j=1}^{N_4} \frac{a_j - ib_j}{p^2 + c_j - id_j}, \tag{7}$$

with (all quantities are expressed in appropriate powers of the unit GeV)

$$\gamma = \frac{13}{22}, \text{ and we choose } Z = 1, \ m^2 \in [2,5], \ \lambda^2 \in [1,4], \ N_1 \in [1,3], \tag{8}$$

$$A_i \in [-0.5, 0.5], \ \{B_i, C_i\} \in [-5, 5], \tag{9}$$

$$\{N_2, N_3\} \in [1,3], \ \{\gamma_i, \tilde{\gamma}_i\} \in [-0.5, 0.5], \ \{\alpha_i, \beta_i, \tilde{\alpha}_i, \tilde{\beta}_i\} \in [1,5], \tag{10}$$

$$\sigma \in [0,1], \ N_4 \in [1,3], \ a_j \in [-1,1], \ b_j \in [0,1], \tag{11}$$

$$c_j \in [0.2, 0.35], \ d_j \in [0.3, 0.75], \ p \in [0, 8.25], \ \omega^2 \in [0.01, 10] \tag{12}$$

and with constraints:

$$D(p^2) \geq 0, \tag{13}$$

$$\ln(\frac{\omega^2 + m^2}{\lambda^2}) > 0, \tag{14}$$

$$B_i \omega^2 + (C_i - \omega^2)^2 \neq 0, \tag{15}$$

$$\begin{aligned}
\lim_{p^2 \to 0^+} \partial_{p^2} D(p^2) &= -\pi \lim_{\omega \to 0^+} \partial_\omega \rho(\omega) + \lim_{p^2 \to 0^+} \partial_{p^2} \left( \sum_{j=1}^{N_4} \frac{a_j + ib_j}{p^2 + c_j + id_j} + \sum_{j=1}^{N_4} \frac{a_j - ib_j}{p^2 + c_j - id_j} \right) \\
&= -\pi \lim_{\omega \to 0^+} \partial_\omega \rho(\omega) - \sum_{j=1}^{N_4} \frac{a_j + ib_j}{(c_j + id_j)^2} - \sum_{j=1}^{N_4} \frac{a_j - ib_j}{(c_j - id_j)^2}.
\end{aligned} \tag{16}$$

The latter constraint is a generalization of the one presented in [29] and can be readily proven along the same lines. Using the residue theorem and following [16], one can also derive yet another constraint that any suitable spectral function should obey

$$\int_\sigma^\infty \rho(\omega) d(\omega^2) + 2 \sum_{j=1}^{N_4} a_j = 0. \tag{17}$$

In general, lattice Monte Carlo propagator data needs proper renormalization. Here we will follow the standard procedure of working in a natural renormalization scheme based on the asymptotic freedom, where we will set the propagators equal to their tree level value at $\mu = 1$ GeV, i.e. we impose the so-called momentum subtraction scheme,

$$D(p^2)\big|_{p^2=\mu^2} = \frac{1}{\mu^2}. \tag{18}$$

The same condition has been imposed on our training data. We chose to uniformly sample each parameter four times in each of their respective ranges and chose $N_1 = N_2 = N_3 = N_4 = 3$. Note that we fixed the amount of poles to three but still allow for less poles by allowing their residues ($a_j$ and $b_j$) to be equal to zero. We then took the Cartesian product of all possible parameter values, removed combinations that did not satisfy the constraints, sampled randomly out of the resulting combinations (depending on the size of the desired data set), and calculated the propagator and spectral density function from the chosen parameters. The spectral function is calculated discretely on 200 uniformly spaced points from $\omega^2 = 0.01$ to $\omega^2 = 10$, while the propagator is calculated on 100 uniformly spaced points between $p = 0$ to $p = 8.25$. 75% of the data went into the training set, 12.5% went into the validation set, and the last 12.5% went into the test set. We chose to use two differently sized training data sets, namely one of size $N = 10,000$ and one of size $N = 30,000$ to examine the impact of the size of the training data on the performance of the neural networks.

## 3 Neural network approach

As non-perturbative lattice QCD propagator data comes from Monte Carlo sampling, there is usually noise affecting the measurement which is not (yet) present in our toy functions. The available experimental samples display a relative error between $10^{-3}$ and $10^{-2}$, with an average of around $0.5 \times 10^{-3}$. Therefore we added Gaussian noise to the raw propagator values with a standard deviation of $0.5 \times 10^{-3}$ before training our neural network, which is more realistic than using exact values and makes our network more robust to noise.

### 3.1 Neural network architecture

Our neural network consists of an input layer, a number of hidden layers and an output layer. The input layer consists of 100 input neurons (for the 100 propagator sample points), while the output layer consists of 213 output neurons (200 for the spectral function sample points, 12 for the 3 possible complex poles and their residues, and one for the IR cutoff). We used batch normalization [30] and Rectified Linear Units (ReLU) [31] between layers in order to make the network more stable and to allow the network to find a non-linear relation between the input and output. The Adam optimizer [32] was used together with a dropout rate of 10% in the training process in order to further improve performance [33]. The mean squared error (MSE) was used as a loss function (Eq. (19)), where the squared difference is calculated between the reconstructed and actual spectral function, both evaluated at $\omega_i^2$, between the reconstructed and the actual poles, and between the reconstructed and actual IR cut-off.

$$\text{MSE} =$$
$$\sum_{i=1}^{200}(\rho(\omega_i^2) - \tilde{\rho}(\omega_i^2))^2 + \sum_{j=1}^{3}\left((a_j - \tilde{a}_j)^2 + (b_j - \tilde{b}_j)^2 + (c_j - \tilde{c}_j)^2 + (d_j - \tilde{d}_j)^2\right) + (\sigma - \tilde{\sigma})^2. \tag{19}$$

Note that other loss functions could also be used, for example by weighting the errors on the poles more heavily or by reconstructing the propagator through Eq. (7) and comparing this to the original propagator. This was not explored in this paper but Kades et al. [24] provide a thorough analysis on three different loss functions, namely spectral function, propagator and parameter loss. To determine how many hidden layers and how many neurons should be in each layer, we trained the network on $N = 10,000$ and $N = 30,000$ training data entries for different combinations of hidden layers and amount of neurons and compared the MSE on the validation set at the end of training. We used a batch size of 100 and trained during 100 and 300 epochs respectively. The results of this analysis can be seen in Fig. 1, where the network with 6 hidden layers and 600 neurons per layer has the lowest validation MSE for $N = 10,000$ and the network with 8 hidden layers and 400 neurons per layer has the lowest MSE for $N = 30,000$. As the latter has a higher MSE than the former, this suggests that the network trained on 30,000 functions is overfitting. Similarly, training the network on less than 10,000 functions reduces performance and underfits the data. We therefore continue the rest of this paper by using the neural network with the lowest validation MSE, namely the one with 6 hidden layers and 600 neurons per layer, which was trained on 10,000 functions [2]. Note that our neural network has only 2 million parameters, compared to e.g. the networks considered in [24] which have at least 41 million parameters, showcasing the efficiency of our model. We used the PyTorch open source machine learning framework [34] to implement our neural networks and ran all our experiments on an Intel i7-8750H CPU clocked at 2.2 GHz and an NVIDIA GTX 1050 GPU.

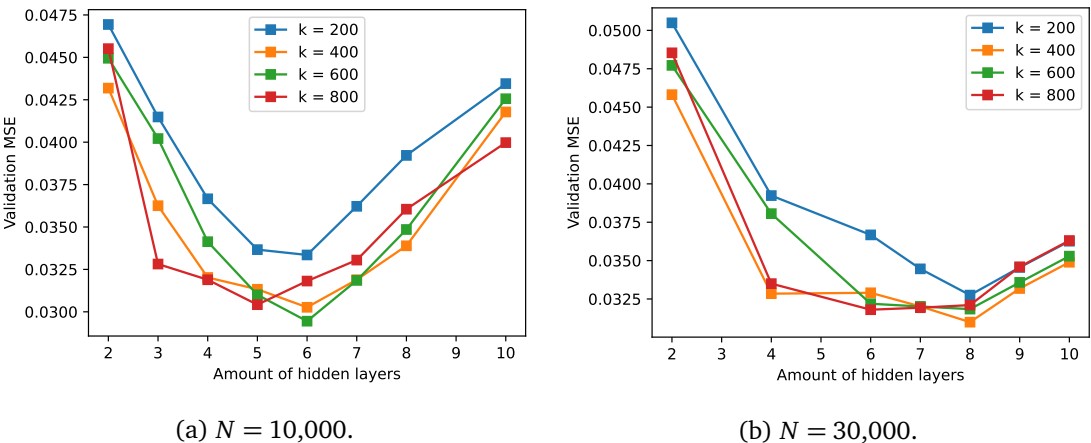

(a) $N = 10,000$.                    (b) $N = 30,000$.

Figure 1: Validation MSE at the end of the training period for different amounts of hidden layers, neurons per layer and training data size N.

# 4 Results and discussion

We will now illustrate the performance of our neural network which was trained on a training set of size $N = 10,000$ and tested on a testing set of size 1,700. Each test propagator was put through the neural network and the reconstruction (consisting of a spectral function, poles and an IR cutoff) was compared to the original values using a normalized variant of the mean absolute error (MAE). All tested propagators were then sorted according to the error of the reconstruction and five reconstructions were plotted, namely the best, the 25[th] percentile, the

---

[2]The used data set, trained neural network and source code is available at
https://github.com/thibaultLe/SpectralANN

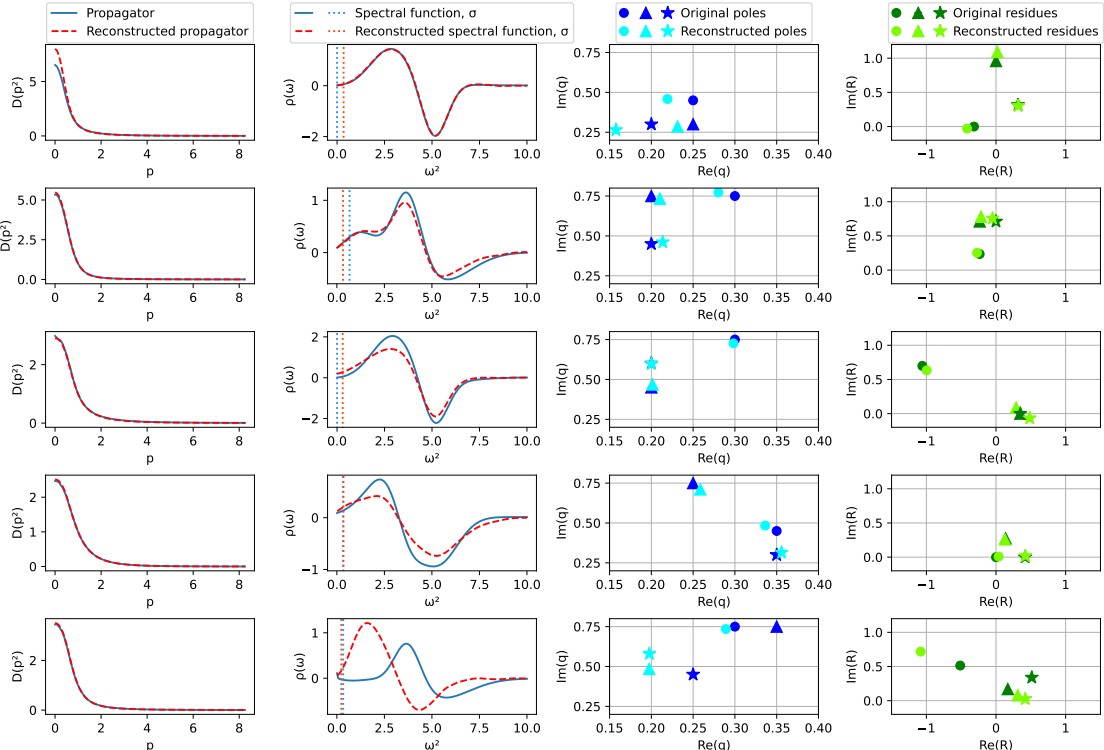

Figure 2: A selection of spectral functions, sorted according to their MAE: the best (top row), the 25th percentile, the median, the 75th percentile, and the worst (bottom row).

median, the 75[th] percentile and the worst reconstruction. The propagator, spectral function, IR cut-off $\sigma$, poles and residues are plotted. This can be seen in Fig. 2. As an additional point of reference, we also plotted the reconstructed propagators, which were calculated by using Eq. (7) and using the reconstructed spectral functions, poles and IR cutoffs as input, but this was not used in the calculation of the error ranking.

The first thing we see in Fig. 2 is that the spectral functions are very well approximated even until the 75th percentile of best reconstructions. The locations of the peaks and troughs match, but the scale is sometimes over- or underestimated. The poles and residues are almost all in the right positions and the reconstructed propagators lean very close to the originals. We can see an interesting facet of the problem in the last (worst) reconstruction: the reconstructed propagator matches the original almost perfectly, yet the reconstructed spectral function and poles do not match their ground truth. This is due to the fact that multiple different combinations of spectral functions and poles can lead to the same propagator. The number of reasonable yet different solutions increases when the noise on the propagator increases, so it is crucial to minimize this noise. Besides comparing the reconstruction to the original function, there is another way of quantifying the quality of our method. We can also analyze whether or not the reconstructions of our neural network satisfy the constraints that we imposed on our training set in Eq. (13), (16) and (17). Of the 1,700 reconstructed propagators, 1644 satisfied constraint (13), only 335 satisfied constraint (16) and 1624 satisfied constraint (17). In total, 311 reconstructions satisfied all constraints. All functions illustrated in Fig. 2 satisfy constraints (13) and (17), but only the second to last one satisfies constraint (16). We cannot conclude that satisfying the constraints is a good predictor of reconstruction accuracy.

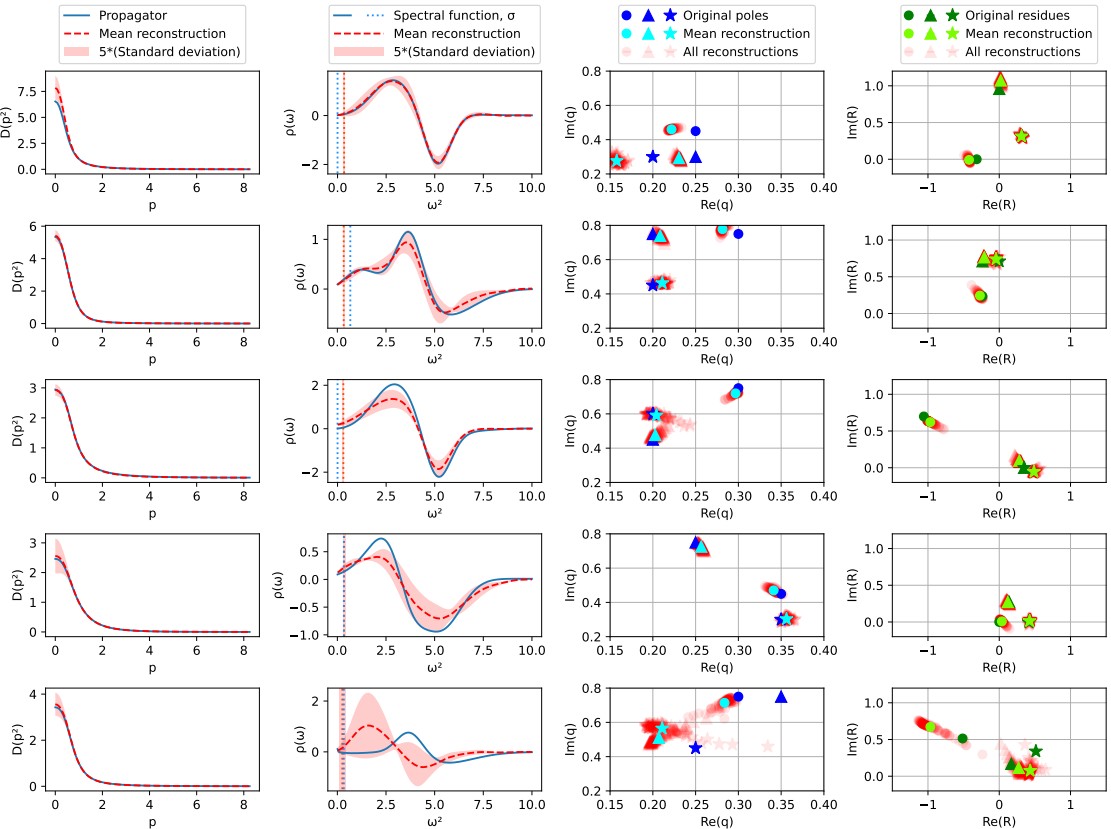

Figure 3: The robustness to noise of the five propagators from the previous figure.

## 4.1 Robustness to noise

Before testing our network on real data, we need to make sure that it is robust to the noise inherent to the Monte Carlo sampling of propagator data. In order to test our network's robustness to this noise, we applied multiplicative Gaussian noise to the raw propagator values from Fig. 2 with a standard deviation of $10^{-2}$, 100 separate times. We then calculated and plotted the mean spectral function reconstruction and the standard deviation around the mean. We did the same thing for the poles and residues, but plotted all reconstructions in light red instead of a standard deviation. The results of this analysis can be seen in Fig. 3. Again as an additional point of reference, we also calculated the propagator for each of these 100 reconstructions and then plotted the mean and standard deviation of these reconstructed propagators. However, because the standard deviations were too small to be noticeable on the figure, we multiplied them by 5.

Fig. 3 shows very promising results, as the mean reconstructions still fit well to the original functions. The standard deviations are small, but larger in the areas that the difference with the original function is larger. This is a useful property as it can be used to quantify the uncertainty of the reconstruction. The poles and residues are also not susceptible to noise because they stay around the mean, which is close to the original (except for the worst reconstructions).

## 4.2 Testing on genuine lattice QCD data

We have demonstrated that our neural network is able to reconstruct spectral functions and poles from our artificially generated dataset, and do so with good robustness to noise. We can now test our network on gluon propagators generated from Monte Carlo sampling [21]. Two configurations were used, namely one with $64^4$ and 20,000 samples, and one with $80^4$

and 18,000 samples. First, linear interpolation was used on the original propagator to match our input format, which makes sure the sample points are 100 uniformly spaced points between $p = 0$ and $p = 8.25$. Next we reconstructed the spectral function, poles and residues for all of the samples in each configuration. From this reconstruction, we again calculated the propagator using Eq. (7). There are no known spectral function, poles or residues for these samples, so we have to rely on the similarity of the reconstructed and the original propagator in order to compare the quality of reconstructions. The reconstructions can be seen in Figure 4 and 5 for the $64^4$ and $80^4$ configurations respectively. As there was not a lot of variation in the reconstructions, only the best, the median, and the worst are illustrated. We see that the reconstructed propagators fit very well to the original, except in the worst cases. The spectral function, poles and residues cannot be compared to their original values, but show good robustness to noise in both configurations. Again we can check how well the reconstructions satisfy the constraints of Eq. (13), (16) and (17). Unlike the artificial test set, all reconstructions satisfied constraints (13) and (17), while only 0.08% and 7.5% satisfied constraint (16) for the $64^4$ and $80^4$ configuration respectively. As for the functions shown in Fig. 4 and 5, they all satisfied constraints (13) and (17), but only the first two in Fig. 5 also satisfied constraint (16). We note that the estimates for the location of the complex conjugate pole masses are consistent with those in e.g. [20, 21].

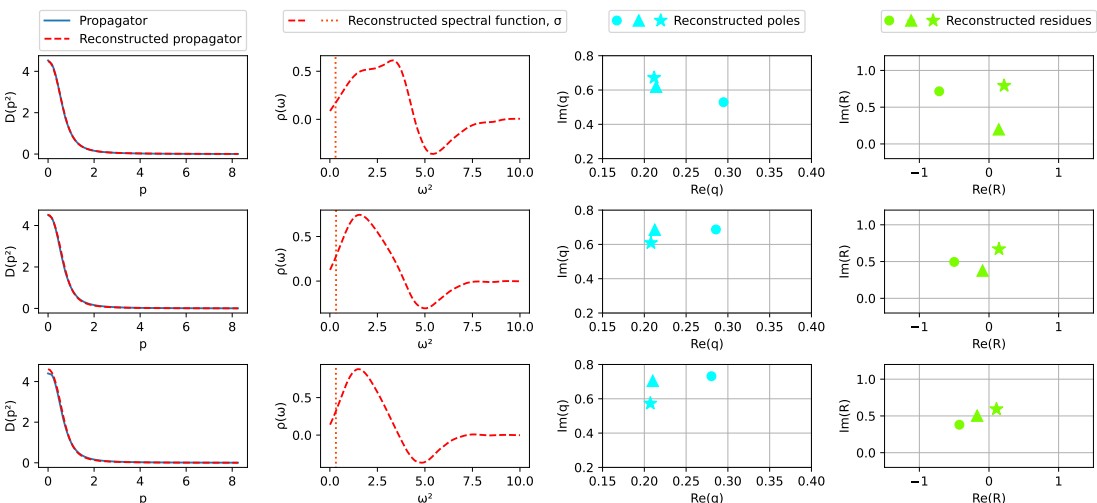

Figure 4: A selection of reconstructed spectral functions from 20,000 gluon propagators generated from Monte Carlo sampling ($64^4$ configuration), sorted according to their MAE: the best (top row), the median, and the worst (bottom row).

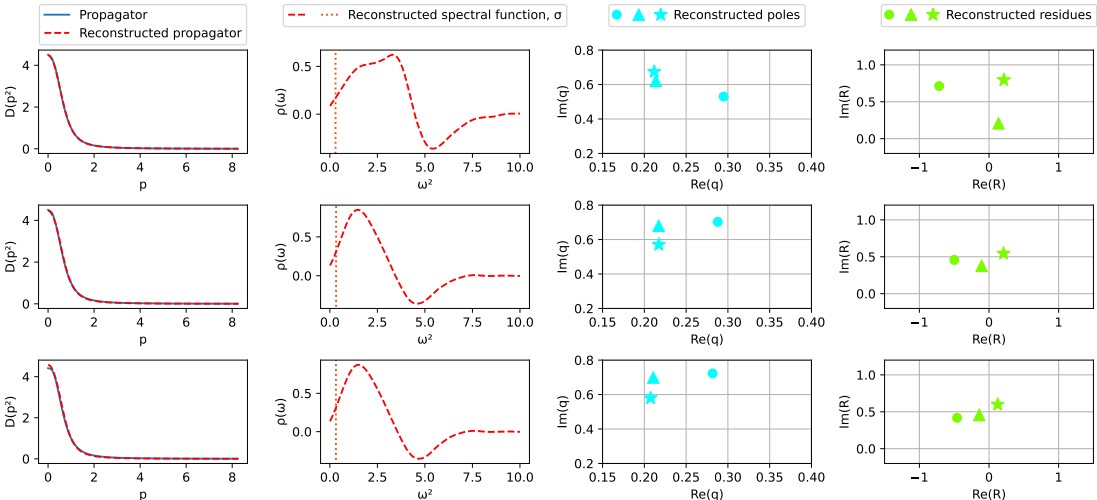

Figure 5: A selection of reconstructed spectral functions from 18,000 gluon propagators generated from Monte Carlo sampling ($80^4$ configuration), sorted according to their MAE: the best (top row), the median, and the worst (bottom row).

# 5 Conclusion

In this paper we have shown that the neural network approach to the analytic continuation problem can be generalized to not only reconstruct non-positive spectral functions, but also pairs of complex poles, their residues and IR cutoffs from propagator data. By using physically motivated toy spectral functions, we generated training data sets and trained a feedforward neural network on these functions. In contrast to alternative approaches [24,35], our network assures that the output spectral function will have the correct UV asymptotics, as dictated by the renormalization group. We examined the performance of different neural network architectures and illustrated their reconstruction accuracy on our data sets with unseen testing data. Most reconstructions were very close to the original functions. The robustness to noise on the propagator was examined and promising results were found. We also applied our network to genuine lattice QCD data for the gluon propagator [21] to further investigate its behaviour in the complex momentum plane. Our findings suggest that a neural network, trained on toy propagators, can recognize if and where pairs of complex poles are present which can be of great importance to expanding our knowledge of confined gluon 2-point functions. Its computation speed is also much faster than traditional methods, as highlighted in [35]. Future work could examine the use of different neural networks (e.g. convolutional neural networks) or using a different loss function (e.g. weighting the loss function). Increasing the scope of the study by allowing for a wider parameter range and an even more diverse training set can also be interesting. Another interesting topic would be to find out if the well-understood confinement-deconfinement transition in compact QED could be connected to a change of the associated spectral density of the photon, using the recent data of [36], perhaps in terms of having, or not, complex conjugate poles. Interesting lattice evidence linking confinement to violations of spectral positivity was presented very recently in the same model, [37].

# Acknowledgments

We thank Orlando Oliveira and Paulo Silva for providing us with the necessary lattice gluon propagator data. We are also grateful to them, Patrick De Causmaecker, and Jorik Jooken for useful discussions.

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
