# Peer review of "Neural network approach to reconstructing spectral functions and complex poles of confined particles"

_SciPost Physics, doi:SciPost Phys. 13, 097 (2022)_

## Round 1 · Referee Report · Anonymous · 2022-6-30

Strengths

1 - Well written
2 - Reproducibility
3 - Useful extension of NN based approaches to the problem of spectral reconstruction

Weaknesses

1 - Interpretation whether cc poles exist in a propagator
2 - Explanation of chosen parameter ranges
3 - Using the fourth constraint

Report

The paper is very well written and easily understandable. Particularly, it excels at begin reproducible with easily readable and runnable code.

My major criticism concerns the parameter ranges chosen in (8)-(10), which seem to be rather small. This goes hand in hand with the conclusion that the NN is capable to decide whether cc poles are present in a correlator. The case of no cc poles is only considered as an edge case in the training data. Additionally, the number of cc pole pairs is fixed to three.
For me, it seems inconclusive if the NN can really differentiate the two cases with the current training data.
Another point concerns the fourth constraint (14). For the spectral function of the gluon this constraint has significant predictive power (negative IR) in the absence of cc poles. Unfortunately, after its introduction, the constraint is not referenced again.

From the plots, it seems like $\sigma=0$ is always favored in the reconstruction, but most likely this is not the case in the training data, maybe the authors would be kind enough to comment on this.

Requested changes

I would suggest adding comments regarding the following two points to the paper:
- Connection between the chosen parameter ranges and if it's sufficient to decide whether cc poles are present
- Discuss the implications and conformity of the fourth constraint with the gluon reconstruction

---

## Round 2 · Author Response

We thank the editor for the opportunity to revise our manuscript. We have carefully considered the reviewer’s comments and changed the manuscript accordingly. Please find below the detailed point-to-point answers to the reviewer’s comments.

We hope that with these changes, our work will be considered for publication in SciPost Physics.

---

## Round 2 · List of Changes

We would like to thank the reviewer for their helpful comments. Please find below our response.

- The parameter ranges from eqs.(8)-(10) were based on estimates from previous work like [*] and our own preliminary experiments which found that larger ranges rarely fit the imposed constraints.
We have added these references and some extra explanation to the paper (Sect. 2.1).

- In the previous version of the manuscript we indeed only considered the case for a fixed number of poles (namely 3). We agree with the Reviewer’s comment. As such, in order to allow the network to not only reconstruct the location of the poles and residues, but also to determine whether poles are present or not, we now included possible residue parameter values of 0 in the training set. When there are less poles than 3, some residue doublets (the a_j and b_j from eq. (7)) will be exactly 0. The goal is that our network only reconstructs the residues for the poles that are present and sets the other residues to zero.
After retraining the network and running all the tests again, we do get promising results: the network is still able to reconstruct the spectral functions, poles and residues with good accuracy, but it is now also able to reject a pole by setting its residue to (approximately) 0. For example, this behavior is visible in the fourth reconstruction of Figure 2 in the revised manuscript.
We have added this explanation and updated all relevant figures in the paper.

- Concerning our usage of the constraints (11), (14) and (15), we believe there was some confusion because we referred to them as the ‘first, second and third constraints’ in the text, while the latter two are actually not the second and third constraints in the numbered equation list of constraints. In order to avoid this confusion in the revised manuscript, we now explicitly refer to the equation numbers instead of their position in the list (see p5, p7).

- For the values of the IR cut-off σ, we have updated the plots to now also include these visually (see e.g. Figures 2 and 3).

References [*]
• D. Binosi, R.-A. Tripolt, Spectral functions of confined particles, Phys. Lett. B 801 (2020) 135171. arXiv:1904.08172, doi:10.1016/j. physletb.2019.135171
• A.F. Falcão, O. Oliveira, P.J. Silva, Analytic structure of the lattice Landau gauge gluon and ghost propagators, Phys. Rev. D 102 (2020) no.11, 114518. arXiv:2008.02614, doi:10.1103/PhysRevD.102.114518
• D. Dudal, O. Oliveira, P.J. Silva, High precision statistical Landau gauge lattice gluon propagator computation vs. the Gribov-Zwanziger approach, Annals Phys. 397 (2018), 351-364. arXiv:1803.02281, doi:10.1016/j.aop.2018.08.019

---

## Editorial Decision

published